# Edible Leafy Vegetables from West Africa (Guinea-Bissau): Consumption, Trade and Food Potential

**DOI:** 10.3390/foods8100493

**Published:** 2019-10-14

**Authors:** Luís Catarino, Maria M. Romeiras, Quintino Bancessi, Daniel Duarte, Diana Faria, Filipa Monteiro, Margarida Moldão

**Affiliations:** 1Centre for Ecology, Evolution and Environmental Changes (cE3c), Faculdade de Ciências, Universidade de Lisboa, Campo Grande, 1749-016 Lisbon, Portugal; lmcatarino@fc.ul.pt (L.C.); fimonteiro@fc.ul.pt (F.M.); 2Linking Landscape, Environment, Agriculture and Food (LEAF), Instituto Superior de Agronomia, Universidade de Lisboa, Tapada da Ajuda, 1349-017 Lisbon, Portugal; dduarte@isa.ulisboa.pt (D.D.); dianafaria@isa.ulisboa.pt (D.F.); mmoldao@isa.ulisboa.pt (M.M.); 3Instituto Nacional de Pesquisa Agrária, C.P. 505 Bissau, Guinea-Bissau; qbancessi@hotmail.com

**Keywords:** African Flora, Wild Edible Plants (WEP), food security, leafy vegetables, nutritional composition

## Abstract

Wild Edible Plants are common in the diet of rural communities of sub-Saharan Africa. In Guinea-Bissau, West Africa, wild plant resources are widely used in human diet, but very few studies have addressed them. The aim of this study is to reveal: (1) the wild and semi-cultivated leafy vegetables consumed in Guinea-Bissau; and (2) the nutritional composition of those plants traded at the largest country market in Bissau. Our results revealed that 24 native or naturalized species with edible leaves are currently consumed by Guinea-Bissau population. Five of them were found at the market: dried leaves of *Adansonia digitata*, *Bombax costatum* and *Sesamum radiatum*, and fresh leaves and shoots of *Amaranthus hybridus* and *Hibiscus sabdariffa*. The analysis of the nutritional properties revealed that leaves contain a significant amount of protein (10.1–21.0 g/100 g, dry basis), high values of macronutrients and micronutrients, as well as of phenolic compounds (13.1–40.3 mg GAE/g) and a considerable antioxidant capacity (DPPH 111.5–681.9 mg Eq Trolox). Although price and availability vary among the leafy vegetables analyzed, these traditional foods appear to be a good dietary component that can contribute to food security in Guinea-Bissau and in other West African countries, as these species are widely distributed in this region.

## 1. Introduction

Wild Edible Plants (WEP) and plant parts such as fruits, roots, flowers and leaves from native and naturalized species are widely consumed across Tropical Africa and deserve greater attention as food sources [1,2]. In West Africa, a large part of the recorded useful plants are actually or potentially consumed by local populations [3,4]. Also, the role of WEP in food safety is potentially important, both as a component in daily diets and as resource food when famine food is scarce [5,6].

The high availability, low price and customary use contribute to the social, economic and nutritional importance of WEP, which are among the most important non-timber forest products traded in African markets. The collection, processing and trade are performed mainly by women, who rely on it as a source of income for households. Therefore, in several countries, the value chain of WEP affects a large number of workers, contributing to the income of their families [7,8].

The properties and nutritional importance of WEP and, in particular, of non-conventional leafy vegetables in the diet of Tropical African populations is highlighted by a large number of publications [9,10]. According to Maundu et al. [11], about 1000 species are used as vegetables in Africa and, for West Africa, Irvine [12] reports at least 150 species of plants with edible leaves, from which about 100 are wild and the remainder are cultivated or semi-cultivated. Dried leaves are widely used as condiments and flavorings for sauces and cooked cereals, sometimes after boiling, sometimes after pounding or drying. Fresh leaves are used as pot-herbs in soups and sauces.

In West Africa, the food situation for many inhabitants is weak, and food security is a concern in several countries, including in rural areas of Guinea-Bissau [13]. The consumption of WEP, namely of leafy vegetables, could contribute to alleviating this problem. Moreover, the populations in rural areas strongly depend on agricultural crops with large inter-annual variations in productivity, as well as (and increasingly) on cashew nut production, which is subject to large fluctuations in market price [14]; WEP could represent a food security resource in times of scarcity [15,16].

A large number of WEP are currently used in the country [17], and two main types of consumption of edible leafy vegetables can be considered: dried and ground leaves (lalos) (Figure 1A–D) and fresh leaves (djambos) (Figure 1E,F). Both lalos and djambos are traded in local markets (*lumus*) and city markets, namely at Bandim, the largest market in the country’s capital, Bissau.

The aim of this work is to report the wild and semi-cultivated leafy vegetables consumed and traded in Guinea-Bissau, to evaluate the nutritional properties of the most important ones sold at Bandim Market, and to discuss their present and potential contribution for food security in the country.

## 2. Materials and Methods

### 2.1. Study Site

Guinea-Bissau is located in West Africa, between 10°59′–12°20′ N and 13°40′–16°43′ W. According to the National Institute of Statistics and Census, in 2014 1,514,451 inhabitants occupied an area of 36,125 km^2^ [18]. Outside the capital (Bissau), the population is mainly rural, and very little in the way of services and infrastructure is available. The climate is tropical, with a wet season from July to November and a dry season from December to June. The main vegetation types in the country are mangroves, palm groves, woodland, savanna woodland and dry forest [19]. Guinea-Bissau’s vascular flora is estimated to include 1524 taxa, 1391 of which are native [17]. The country comprises a diverse mosaic of about 30 different ethnic groups of Muslim or African traditional beliefs, with a deep knowledge of the use of natural resources. A recent study revealed that about 15% of vascular plant species are used in traditional medicine [20]. The Bandim Market is the largest food market in the country and is located at the capital, Bissau; a wide range of products is sold there, including traditional foods and medicines coming from all over the country.

### 2.2. Characterization of Species with Leaves Consumed in Guinea-Bissau

After extracting a first list of candidate species from Catarino et al. [17], a prospection was made at LISC herbarium, from Lisbon University, to retrieve the available information on the plants with edible leaves recorded in Guinea-Bissau’s flora. LISC herbarium presently holds the largest collection of vascular plants from the country, including recent vouchers obtained during ethnobotanical studies. Plant names follow The Plant List nomenclature (www.theplantlist.org).

The characterization of the species with edible leaves was obtained by consulting the references found in herbarium vouchers, namely about the type of consumption, as well as the information compiled by Catarino et al. [17] concerning uses and vernacular names. A representative voucher was selected for each species.

### 2.3. Prospection and Sampling of Marketed Species

Surveys were made at Bandim market in June and July 2017 and in May 2019 to record the fresh and dried wild and semi-cultivated leafy vegetables available, to collect data on their prices, and to obtain samples for analyzes. For the five leafy vegetables sold at this market (see below), detailed information concerning species ecology, types of products marketed, production time and availability was obtained both at the market and from the LISC Herbarium.

The price and units of sale of each product were asked for from several sellers, and the corresponding portions were weighed. For the present study, the price was established after five constant prices and weight units were recorded. Samples of each product: dried and ground leaves (lalos) of *Adansonia digitata, Bombax costatum* and *Sesamum radiatum* (see Figure 1A–D), and fresh leaves (djambos) of *Amaranthus hybridus* and *Hibiscus sabdariffa* (Figure 1E,F) were acquired at Bandim Market from different sellers, mixed and transported to the Laboratory of the Institute of Agronomy (ISA, University of Lisbon), where their physico-chemical properties were analyzed.

### 2.4. Physico-Chemical Characterization of Lalos and Djambos

Physico-chemical determinations were conducted in triplicate. Fresh leaves (djambos) were lyophilized prior to chemical analysis.

Moisture content—Moisture content was evaluated according to a standard gravimetric method [21].

Water activity (aw)—Water activity was measured using a water activity meter (Rotronic HygroPalm, HP23-AW-A, Bassersdorf, Switzerland) at 20 ± 0.1 °C.

Protein content—For the determination of nitrogen content, the Kjeldahl method was used [22]. The total nitrogen content was multiplied by 6.38 to determine total (crude, total N × 6.38) protein, expressed in g/100 g of dry weight.

Lipid content—The total lipid content was determined by the Soxhlet extraction with hexane as solvent.

Mineral analysis—Concentrations of minerals in lalos and djambos were determined by inductively coupled plasma optical emission spectrometry (Thermoscientific iCAP7000 series ICP spectrometer, Waltham, MA, USA) after digestion (in aqua regia, a mixture of nitric acid and hydrochloric acid (1:3) at 105 °C during 60 min).

#### Antioxidant Properties of Lalos and Djambos

Extracts of lalos and djambos—Extracts from lyophilized fresh (djambos) and dried (lalos) leaves were obtained by mixing 1 g of powder with 40 mL of a methanol–water solution (50% each) and stirred with an Ultra Turrax^®^ homogenizer (IKA Ultra Turrax digital, Model T25 basics, Staukfen, Germany) for 1 min at 13,500 rpm. The overall mixture was stored in the dark at 25 °C for 60 min and centrifuged (25,000× *g* for 10 min, Hermle Labortechnik, Model Z 383 k, Wehingen, Germany). The supernatant was separated, and the pellet was extracted again with an acetone–water solution (70% acetone) using the same methodology described above. Finally, the recovered supernatants were mixed and stored in the same flask, in the dark.

Ferric reducing antioxidant power—The ferric reducing antioxidant power (FRAP) method was performed according to Pulido et al. [23]. A 90 μL aliquot of the extracts, obtained as described in the previous section, was transferred to glass tubes, added up with 270 µL of distilled water and mixed with 2.7 mL FRAP reagent, stirred in a vortex and maintained in a water bath at 37 °C. After 30 min of reaction, the absorbance at 595 nm was measured using a spectrophotometer (UNICAM UV/Vis Spectrometer UV4, Cambridge, UK). The spectrophotometer was calibrated with FRAP reagent. The FRAP results were expressed as µmol FeSO4.7H2O equivalents/g dry matter or dry film, using a standard curve of FeSO_4_.7H_2_O (mM) as reference. Determinations of ferric reducing antioxidant activity were performed in triplicate.

DPPH radical scavenging capacity—The methodology recommended by Ferreira et al. [24] was used for determination of antioxidant activity by 2,2-diphenyl-1-picrylhydrazyl (DPPH) method. Lyophilized extracts of lalos and djambos (100 μL) were mixed with 3900 μL of DPPH solution (0.06 mM), and the reaction lasted 40 min at room temperature in the dark. The reduction of DPPH radical in the solutions was evaluated by measuring the absorbance at 515 nm using a spectrophotometer (UNICAM UV/Vis Spectrometer UV4, Cambridge, UK) [25]. The results were expressed as µmol Trolox equivalents (TE)/g dry mater, based on a standard curve of Trolox (6-hydroxy-2,5,7,8-tetramethylchroman-2-carboxylic acid from Aldrich, Beijing, China). The tests were performed in triplicate.

Total phenolic content—Total phenolic content was determined by spectrophotometry according to the method of Ribéreau-Gayon [26] by measuring the absorbance at 280 nm in a quartz cell of 1 cm optical path (UNICAM UV/Vis Spectrometer UV4, Cambridge, UK). The results were expressed in mg of gallic acid equivalents per g dry weight (mg GAE/g), based on a calibration curve derived by linear regression, established from concentrations of 0 to 40 mg L^−1^ of gallic acid.

## 3. Results

### 3.1. Species with Edible Leaves

A total of 24 wild or semi-cultivated plant species with edible leaves were found in the flora of Guinea-Bissau with data confirming their consumption in the country (Table 1), belonging to 20 genera and 13 families, all dicotyledons. Most of them are from the families Malvaceae, with seven species, and Amaranthaceae, with four species; Apocynaceae, Convolvulaceae and Pedaliaceae are the other represented families, with two species each. Fifteen species grow as herbs, five as trees and four as vines.

Concerning edibility, the information in the herbarium vouchers indicated 15 species with “Leaves cooked as vegetable”, meaning that the fresh leaves are used as an ingredient in cooked meals. Four species had herbarium references to the use of leaves as flavoring, in sauces, or as lalo. For these types of uses, small quantities of plant material are needed. For the remaining five species, there was only a reference to the edibility of leaves.

For some species, the vernacular names reflect their use as food, such as *Ficus dicranostyla*, known as d’jambô or djambo-surei, in fulani, and *Sesamum radiatum*, known as lalo-caminho in Guinea-Bissau creole.

### 3.2. Prices and Types of Uses

Five species of wild or semi-cultivated leafy vegetables were found currently marketed at Bandim Market: dried leaves (lalos) of the trees *Adansonia digitata* and *Bombax costatum*, and of the herb *Sesamum radiatum*, and fresh leaves and shoots (djambos) of the herbs *Amaranthus hybridus* and *Hibiscus sabdariffa* (Table 2).

The selling price for lalos and djambos was quite different. The dried and crushed leaves of *Adansonia digitata, Bombax costatum* and *Sesamum radiatum*, with prices of 1470, 2014, and 1760 XOF per kg, respectively (CFA Franc, the local currency, with the international code XOF), are expensive for most of the local consumers. However, these are dried and processed products and their price can be at least partially explained by the processing and storage requirements. Accordingly, lalos are mainly used in small quantities, in sauces or as flavoring ingredients. Conversely, the djambos of *Amaranthus hybridus* and *Hibiscus sabdariffa* are comparatively inexpensive (62 and 151 XOF per kg, respectively), even when considering that they are fresh and unprocessed leaves, and used in large quantities, as pot vegetables.

### 3.3. Physico-Chemical Characterization of the Analyzed Products

The physico-chemical characterization of the analyzed lalos and djambos is shown in Table 3, Table 4 and Table 5. The *Adansonia digitata, Bombax costatum* and *Sesamum radiatum* lalos had moisture contents ranging from 7.0 to 10.6%, and water activities between 0.31 and 0.62, which places them in the dehydrated products category [27]. These values of water activity are considered inhibitory of microbial development and, therefore, they allow the stabilization of the products. Microbial growth only occurs at aw values exceeding 0.6 (some fungi and yeasts) and over 0.90 (most bacteria) [27]. Djambos from *Amaranthus hybridus* and *Hibiscus sabdariffa* consist of fresh leaves with a moisture content of 84.2% and 83.0%, and are therefore suitable products for sale when fresh.

All lalos and djambos presented protein contents above 10 g/100 g, and the one from *Amaranthus hybridus* showed the highest value, 21.0 g/100 g dw, which is considered high for vegetable products. This composition makes these products an important protein source for these populations. The lipid contents ranged from 1.7 to 3.2 g/100 g dw.

Regarding its mineral content, analyzed lalos and djambos presented high values of macronutrients and micronutrients, namely calcium (1478.7 to 2751.7 mg/100 g dw), phosphorus (189.7 to 478.7 mg/100 g dw), iron (33.9 to 83.7, mg/100 g dw) and magnesium (667.9 to 1013.5 mg/100 g dw). The djambo from *Amaranthus hybridus* presents the highest values, on a dry basis, for most of the mineral elements analyzed. These high levels of minerals in non-fertilized crops could be related to soil and climate issues that should be further studied.

To understand the dietary impact of the consumption of these traditional foods, the percentages of Daily Recommended Doses (DRDs) [28] for some elements provided by lalos and djambos were estimated. It was calculated the percentage of DRDs that would be compensated by a daily intake of 2 tablespoons of lalos (≅ 15 g), or about 80 g of djambos, which is the estimated average consumption by the population of Guinea-Bissau. The analyses of Figure 2 make it possible to conclude that intake of two tablespoons daily of lalos from *Adansonia digitata, Bombax costatum* and *Sesamum radiatum* or about 80 g of djambos from *Amaranthus hybridus* and *Hibiscus sabdariffa* corresponds to about one third of the DRD of calcium and magnesium, about 100 percent of iron, and 170–290 percent of the DRD of manganese. Excess manganese may hinder iron adsorption and doses of iron above 20 mg per day may cause stomach disorders [29,30].

The analyzed samples showed a high phenolic content of 34.4 to 40.3 mg gallic acid equivalent (GAE)/g dry weight for lalos and 13.1 to 18.8 mg GAE/g for djambos. The obtained results are in agreement with those observed by Lutz et al. [31] for fresh and dehydrated spinach (10.6 and 51.4 mg GAE/g, respectively). Spinach is one of the vegetables that presents the highest phenolic content [32].

Lalos showed consistently higher antioxidant capacities than djambos in the three samples analyzed (Table 5), with DPPH values of 526.6 to 681.9 µmol TE/g dw for lalos and 111.5 to 180.8 µmol TE/g dw for djambos. The antioxidant capacity assessed by FRAP method indicated a strong agreement with DPPH method. These results were expected considering the phenolic content of the samples. The results for lalos and djambos are in accordance to those observed by Lutz et al. [31] for fresh and dehydrated spinach (51.4 and 537.1 µmol TE/g dw, respectively).

The inclusion of lalos and djambos in the diet can bring about additional health benefits due to their high levels of biologically active compounds (minerals and phenols) and due to their powerful antioxidant capacity, which are known to be highly beneficial to health.

## 4. Discussion

Food security is a concern in Guinea-Bissau, and our study highlights the importance of WEP consumption by local populations, which are strongly dependent on agricultural crops that have large inter-annual fluctuations [14,15]. Our research revealed that only 5 of the 24 species with reported edible leaves in Guinea-Bissau were found to be currently marketed. Several reasons may be presented to explain this. Some edible leaves are not suitable for drying and grinding or even to be sold fresh, and can therefore be used only locally. Also, given the low consumption of vegetables, probably there is not enough demand for other leafy WEP. On the other hand, a fair number of households have access to leafy vegetables from other sources, namely home gardens, even in the main cities. Finally, it is worth noting that our survey at Bandim market was made during three months in the dry season, and it is possible that other species are traded in other periods of the year, namely fresh leafy vegetables.

Concerning the five species found at Bandim market, the use of leaves, fruits and flowers of *Adansonia digitata, Amaranthus hybridus, Hibiscus sabdariffa* and *Sesamum radiatum* is reported in West Africa and elsewhere, and their composition and dietary characteristics have been analyzed by several authors [9,10,12,33,34,35,36,37,38]. *Bombax costatum* is also referenced in some studies, but the compositional data is only available for flowers and young fruits [12,39,40,41]. *Amaranthus* species are among the most important leafy vegetables in Africa [42]. This genus presents some taxonomical complexity and the species from the complex *Amaranthus cruentus—Amaranthus hybridus* are cultivated or semi-cultivated in several countries and also used for medicinal and phytochemical purposes [4,34].

In Guinea-Bissau, the way the leafy vegetables are consumed varies: lalos are used in small quantities as flavoring, while djambos are used in larger quantities, in soups or stews, replacing cabbage. The availability of these products throughout the year is relatively constant: lalos are dried and ground during the species’ leaf production season and preserved throughout the year, while djambos’ species can be produced during the year as they are cultivated, thus watered, and consumed as fresh leaves.

The use of leaves from the two Malvaceae trees (i.e., *Adansonia digitata* and *Bombax costatum*) found at Bandim market has already been documented [43]. According to Lykke et al. [40], leaves from indigenous trees in Burkina Faso, such *as Bombax* and *Adansonia*, are frequently collected from fields or bush and eaten fresh, but they are dried and stored for use throughout the year as well. Irvine [12] refers that dried leaves are widely used as condiments and flavorings for sauces and cooked cereals, namely from *Adansonia digitata*. The compositional data obtained for this species confirm the nutritional value of its leaves, which Sena et al. [44] consider nutritionally superior to the fruits.

As far as we know, there were no available data on the chemical composition of *Bombax costatum* leaves. Our data for this species showed affinities with those of *Adansonia digitata*, with comparable contents of protein but some significant differences in nutrient content. For *Amaranthus hybridus* the results of this investigation are in line with previous works and point to a valuable source of protein as well as macro and micronutrients (e.g., [34]).

Regarding phenolic contents, results showed that leaves present much higher levels (13.1 to 36.0 mg GAE)/g dw) than those observed by Gunathilake and Ranaweera [45] in fresh leafy vegetable samples of *Amaranthus lividus* and *Amaranthus caudatus*, collected from various places in Sri Lanka: 3.25 and 2.37 mg GAE/g dw, respectively. The highest value reported by those authors, among 34 types of fresh leafy vegetable samples, was 11.03 mg GAE/g dw for *Sesbania grandiflora* leaves. The contribution of the edible leafy vegetables to the intake of polyphenols and antioxidant capacity in through the diet of Guinea-Bissau population is potentially important, namely with the inclusion of lalos.

High levels of several mineral elements, namely K, Ca, Mg, P, Fe, were observed in both djambos and lalos screened. Elevated levels of calcium in *Adansonia digitata* leaves (307 to 2640 mg/100 g dw) have also been reported [43], which is in accordance to our data. In a previous study, Leterme et al. [46] analyzed the mineral content of 68 species from the tropical rainforest of Colombia, with several species displaying high contents of potassium (1.782 mg/100 g dw), calcium (280–1242 mg Ca/100 g edible portion) and iron (0.7–8.4 mg Fe/100 g edible portion). In fact, this study reported for the first time the highest content of Ca on an African tree leaves (*Trichanthera* sp., Acanthaceae), ca. 62 g Ca/kg DM, used in human nutrition. Considering the high mineral contents on lalos and djambos analyzed, one can recognized the great potential of these WEP to play a major role in a more sustainable and diversified diet, considering they can be regarded as good vegetable sources of many mineral elements.

To understand the dietary impact of the consumption of these traditional foods, the percentage of DRDs [28] for some elements provided by lalos was estimated. As shown in Figure 2, two tablespoons of lalo meets the daily requirements of some elements, namely calcium and magnesium, while exceeding elements such as manganese and iron. Excess of manganese may hinder iron adsorption and doses of iron above 20 mg per day may cause stomach disorders [29,30].

In Guinea-Bissau, rice is the staple food, and leafy vegetables are generally used in small quantities. A low consumption of vegetables by West African populations has already been noted by authors such as Busson [3], who refers the almost exclusively carbohydrate-based nutrition. Also, according to Yang and Keding [47], vegetables are usually considered a relish or even a condiment or spice, and always as a side dish accompanying other (usually starchy) food, and the amount of vegetables consumed can be rather small.

## 5. Conclusions

Considering the nutritional properties and the wide availability of the studied leafy vegetables, these traditional foods can be considered an alternative dietary component to contribute to food security in Guinea-Bissau. In addition to their nutritional value, these products presented high levels of phytochemicals, phenols, and powerful antioxidant capacities.

Classified as “dehydrated products”, lalos are stable, as far as microbial risk is concerned, proving to be useful forms for trading and consuming leafy vegetables. The leaves, produced during the wet season, are dried and crushed at the beginning of the dry season and sold until the end of that period. This annual cycle of production, trade and consumption of lalos ensures the availability of some vegetables during the dry season, when fresh ones are scarce. On the other hand, the fresh leaves or djambos, are more tailored for immediate consumption and have lower prices which permit the use of larger quantities.

As only a small part of the WEP recorded in the country’s flora are found as traded, it seems that there is some space to the increase of trade and consumption of more species. The study of the properties and composition of the WEP and the encouragement of a more intensive consumption of the most useful species can contribute to improve the dietary condition of the Guineans, both in rural and urban areas. This would greatly benefit from further studies, linking the production capacity, the market chain characteristics and the promotion of vegetable consumption.

## Figures and Tables

**Figure 1 foods-08-00493-f001:**
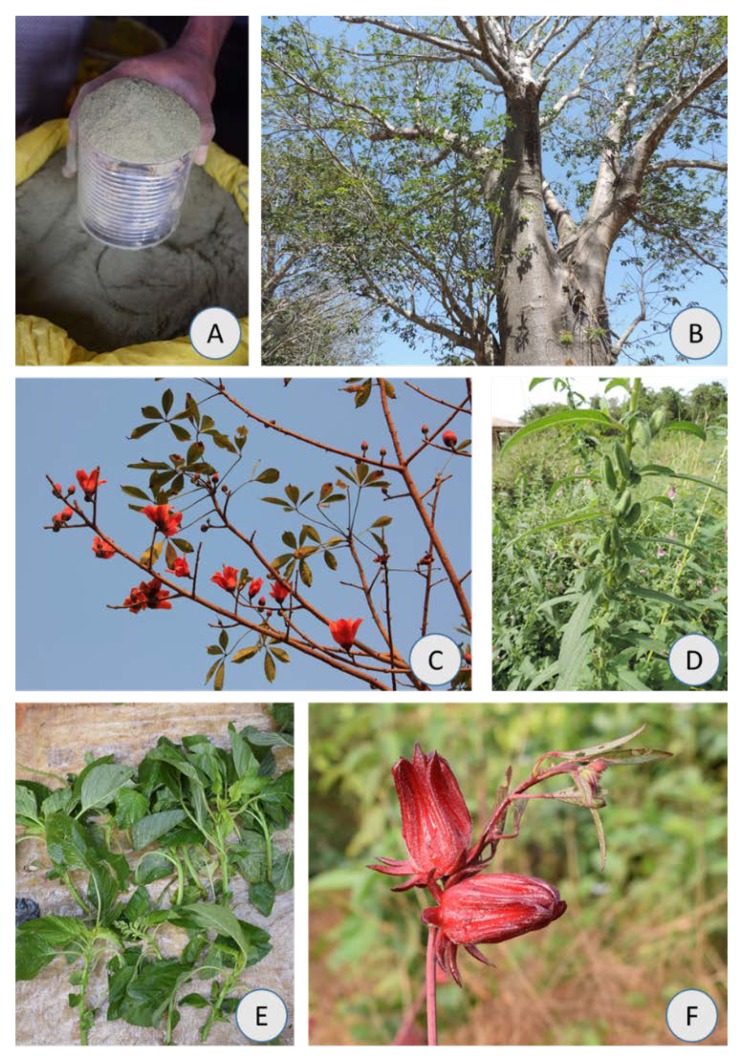
Two main types of consumption of edible leafy vegetables. (**A**–**D**) Dried and ground leaves, lalos, and (**E**–**F**) fresh leaves, djambos. Both lalos and djambos are traded in local markets, such as at Bandim, Bissau. Lalos: *Adansonia digitata* (**A**,**B**), *Bombax costatum* (**C**); and *Sesamum radiatum* (**D**). Djambos: *Amaranthus hybridus* (**E**) and *Hibiscus sabdariffa* (**F**). Photos made by L. Catarino.

**Figure 2 foods-08-00493-f002:**
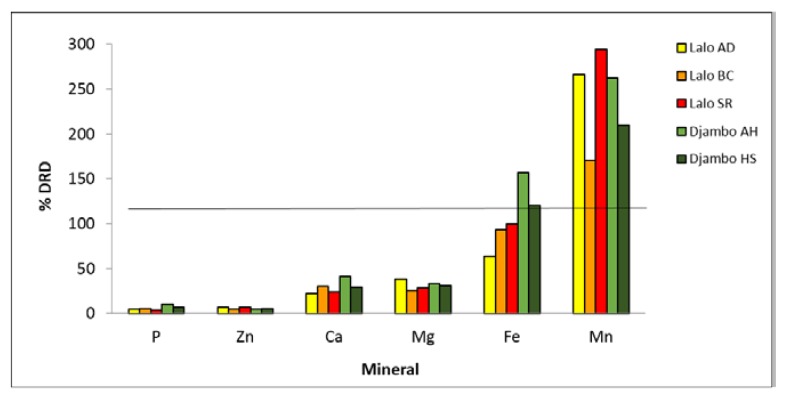
Percentage of the Daily Recommended Doses (DRD) supplied by two daily tablespoons of lalos from *Adansonia digitata* (AD), *Bombax costatum* (BC) and *Sesamum radiatum* (SR), (c. 15 g) or about 80 g of djambos from *Amaranthus hybridus* (AH) and *Hibiscus sabdariffa* (HS).

**Table 1 foods-08-00493-t001:** Data on species with leaves consumed in Guinea-Bissau.

Family	Species *	Vernacular Names **	Habit	Edibility of Leaves	LISC Voucher and Year
Acantaceae	*Nelsonia canescens* (Lam.) Spreng.	n’tobetobe, untúb-túbè (ba); dêpê-farró (fu); nhicicumbalium (md)	Herb	Cooked as vegetable	Vidigal et al. 49; 1991
Aizoaceae	*Seszuvium portulacastrum* (L.) L.	n’bossé (nl); uondgi (ss); bossaha, burunquè (ba)	Herb	Cooked as vegetable	Diniz & Gonçalves 1777, 1997
Amaranthaceae	*Alternanthera sessilis* (L.) R.Br. ex DC. (Syn.: *A. nodiflora* R.Br.)	bocha (ba)	Herb	Cooked as vegetable	Diniz et al. 390, 1990
Amaranthaceae	***Amaranthus hybridus*** **L.**	brêdo-fêmea (cr)	Herb	Fresh, used as vegetable	Pereira 3200, 1962
Amaranthaceae	*Amaranthus spinosus* L.	boro-boro, djambo (fu); brêdo, bride (cr)	Herb	Cooked as vegetable	Vidigal et al. 52, 1991
Amaranthaceae	*Amaranthus viridis* L.	brêdo (cr); bóròbórò (fl)	Herb	Cooked as vegetable	Alves Pereira 2517, 1961
Apocynaceae	*Leptadenia lancifolia* (Schumach. & Thonn.) Decne. (Syn.: *L. hastata* (Pers.) Vatke.)	inrokdé, n’rocdè, enrocodé (ba); cibode (cr), djambo-soredjé, safaro, safarodje (fu); bé-thácare (mc); bissacra (pp)	Vine	Young leaves and flowers cooked as vegetable	Diniz & Gonçalves 1823, 1997
Apocynaceae	*Tacazzea apiculata* Oliv.	nhandurrabo (bj); sapaté (bf); sapátè (cr); saparô, mancahaneidje (fu)	Vine	Cooked as vegetable	Martins & Catarino 1232, 1996
Convolvulaceae	*Ipomoea aquatica* Forssk.	intambelata (ba); quelô (ff); djambo (= edible leaf) (md); bole-bola (nl)	Herb	Cooked as vegetable	Diniz et al. 542, 1990
Convolvulaceae	*Merremia tridentata* (L.) Hallier f.	lata (ba)	Vine	Cooked as vegetable	Martins & Catarino 1267, 1996
Dillenaceae	*Tetracera potatoria* Afzel. ex G.Don	ebirito (bj); n’átá (nl)	Vine	Used in sauces.	Moreira 250, 1994
Lamiaceae	*Platostoma africanum* P.Beauv.		Herb	Used as flavoring	Gonçalves et al. 90, 1988
Malvaceae	***Adansonia digitata*** **L.**	late (ba); uáto (bj); cabacera, calabacera (cr); boé (fu); bedom-hal, burungule-burunque (mc); bebáque, brungal (mj); burungule (pp); kiri (ss)	Tree	Dried, as lalo	Espírito Santo 1191, 1938
Malvaceae	***Bombax costatum*** **Pellegr. & Vuillet**	buúforè, bumbum (ba); polóm-fidalgo, polóm-fôro, sumauma (cr); luncum, djóia, djóè (fu); djóia, djóè (ff); belofa (mc); buncum-ô (md); belofa (mj); ulófo (pp)	Tree	Dried leaves and flowers, as lalo	Catarino 2394, 2016
Malvaceae	*Chorchorus aestuans* L.	cunhunho (bf); lalel-bábos (fu)	Herb	Edible	Espírito Santo 1005, 1937
Malvaceae	***Hibiscus sabdariffa*** **L.**	n’batú, umbatú (ba); busságá (plant), n’tchága (leaves) (bf); baguiche, baguitche, bajique (cr); fólerè (fu); cutchá (md); uncuanto (mj); otésse (pp)	Herb	Edible	Diniz et al. 682, 1991
Malvaceae	*Hibiscus surattensis* L.	m’datu, m’bat’u (ba); baguitch-di-mato, bajique-do-mato (cr); conisanto (ss)	Herb	Edible	Vidigal et al. 279, 1995
Malvaceae	*Melochia corchorifolia* L.	tobre-guelonguê, sôre (fu); cumaré-turo (md)	Herb	Edible	Martins et al. 97, 1989
Malvaceae	*Sterculia tragacantha* Lindl.	n’bama, umbama (mj); umbufúrè, búè (ba); eritô (bj); úcud, dácud (cb); nassino, pau-corda, pau-de-saia (cr); tchapelêguê (fu); bamé (mc); atakssulé (td)	Tree	Cooked as vegetable	Diniz & Gonçalves 2162, 1997
Moraceae	*Ficus dicranostyla* Mildbr.	sur (ba); suredje, surei, djambo-surei, d’jambô (fu); anak (td)	Tree	Cooked as vegetable	Martins & Catarino 1219, 1996
Myristcaceae	*Pycnanthus angolensis* (Welw.) Warb.	súngala (fu); menebantam-ô (md)	Tree	Cooked as vegetable	Martins & Moreira 881, 1995
Nyctaginaceae	*Boerhavia erecta* L.	sabi-cura, fendala, cumara-sabi (fu)	Herb	Edible	Vidigal et al. 54, 1991
Pedaliaceae	*Sesamum indicum* L.	bene (fu)	Herb	Cooked as vegetable	Diniz et al. 1312, 1995
Pedaliaceae	***Sesamum radiatum*** **Schumach. & Thonn.**	tchaba-laba (ba); lalo-caminho (cr)	Herb	Cooked as vegetable	Sane 185, 1988

* Species in bold were analyzed in this study. ** Key for vernacular names’ languages: ba—balanta; bf—beafada; bj—bijago; cb—cobiana; cr—creole; fl—felupe; fu—fula; ff—futa-fula; mc—mancanha; md—mandinga; mj—manjaco; nl—nalu; pp—papel; ss—sosso; td—tanda.

**Table 2 foods-08-00493-t002:** Characteristics of the products and species analyzed.

Products/Species	Product Type and Use	Production Time	Selling Unit and Price in XOF *	Mean Weight **	Price Per Kg in XOF, EUR and USD *	Plant Characteristics
**Lalos**						
*Adansonia digitata*	Dried and crushed leaves are used as ingredient in stews as substitute of okra, or quiabo (*Abelmoschus esculentus*).	Produced in the dry season (February to April), when leaves are easily dried. Marketed throughout the year.	Cup of 1 L/500 XOF	340.2 g	1470 XOF (2.24 €; $2.76)	Large tree, deciduous, mostly grown near villages. The fruits are edible and largely traded in the markets. Young leaves are used to make lalo.
*Bombax costatum*	Dried and crushed leaves are used as ingredient in stews as substitute of okra, or quiabo (*Abelmoschus esculentus*).	Produced at the end of the dry season and beginning of the rains (April to July), period of vegetative regeneration of the tree. Marketed throughout the year.	Cup of 1 L/500 XOF	248.3 g	2014 XOF (3.07 €; $3.77)	Medium to large tree, deciduous, in savannah woodland. The petals are also edible, and the wood is used to make artifacts and furniture. Young leaves, as well as flower petals, are used to make lalo.
*Sesamum radiatum*	Dried and crushed leaves are used as ingredient in stews.	Produced in home gardens at the beginning of the dry season; marketed during the dry season.	Cup of 1 L/500 XOF	284.0 g	1760 XOF (2.68 €; $3.30)	Annual herb, semi-cultivated or cultivated. Both leaves and seeds are edible; the seeds are sold as cash crop.
**Djambos**						
*Amaranthus hybridus*	Fresh leaves and branches are used as vegetables in soups or stews, replacing cabbage.	Produced in home gardens throughout the year; marketed mainly during the dry season.	Bundle of 25 branches/25 XOF	403.0 g	62 XOF (0.09 €; $0.12)	Annual herb, semi-cultivated or cultivated. Both leaves and seeds are edible, but seeds do not appear to be much consumed.
*Hibiscus sabdariffa*	Fresh leaves are used as vegetables in soups or stews.	Produced in home gardens throughout the year; marketed mainly during the dry season.	Bundle of 5 branches/25 XOF	166.4 g	151 XOF (0.23 €; $0.28)	Annual herb, semi-cultivated or cultivated. Leaves, fruits and petals are edible. The fruits are used as legume and the petals to make juice.

* The local currency is the CFA Franc, with the international code XOF; with the exchange rate of 1 EUR = 655.956 XOF and 1 USD = 533.554 XOF. ** Average of 5 samples.

**Table 3 foods-08-00493-t003:** Physico-chemical characterization of products and species.

Products/Species	Water Activity (a_w_)	Moisture Content (%)	Lipids (g/100 g dw)	Protein (g/100 g dw)
**Lalos**				
*Adansonia digitata*	0.62	10.6 ± 0.2	3.2 ± 0.7	10.1 ± 0.7
*Bombax costatum*	0.61	9.8 ± 0.2	2.4 ± 0.2	10.8 ± 0.9
*Sesamum radiatum*	0.31	7.0 ± 0.4	3.1 + 0.5	13.3 + 0.1
**Djambos**				
*Amaranthus hybridus*	0.78	84.2 ± 0.6	1.7 ± 0.1	21.0 ± 1.0
*Hibiscus sabdariffa*	0.98	83.0 ± 0.1	2.3 + 0.6	13.7 ± 0.1

Species in bold were analyzed in this study. Means of three replicates ± SD.

**Table 4 foods-08-00493-t004:** Mineral composition (mg/100 g dw) of products and species.

Products/Species	Na	K	Ca	Mg	P	S	Fe	Cu	Zn	Mn	B
**Lalos**											
*Adansonia digitata*	15.1	686.6	1478.7	1013.5	217.3	248.4	33.9	1.4	4.2	35.5	3.7
*Bombax costatum*	47.9	718.0	2002.5	667.9	265.8	208.6	49.6	1.2	2.8	22.8	3.2
*Sesamum radiatum*	13.2	976.2	1569.7	757.2	189.7	180.2.	53.1	1.7	4.3	39.2	1.9
**Djambos**											
*Amaranthus hybridus*	34.6	2855.7	2751.7	884.8	478.7	412.9	83.7	0.8	2.7	35.0	4.2
*Hibiscus sabdariffa*	6.5	178.2	1975.1	829.9	340.6	291.5	64.2	1.1	3.4	28.0	7.2

Species in bold were analyzed in this study. Means of three replicates.

**Table 5 foods-08-00493-t005:** Antioxidant capacity of products and species.

Products/Species	Total Phenolic Content (mg GAE/g dw)	FRAP (µmol/g dw)	DPPH (µmol TE/g dw)
**Lalos**			
*Adansonia digitata*	34.4 ± 1.3	416.5 ± 28.7	580.0 ± 46.8
*Bombax costatum*	36.0 ± 0.5	396.9 ± 18.9	681.9 ± 15.3
*Sesamum radiatum*	40.3 ± 0.9	662.2 ± 15.9	526.6 ± 50.2
**Djambos**			
*Amaranthus hybridus*	18.8 ± 0.2	266.7 ± 48.0	180.8 ± 25.0
*Hibiscus sabdariffa*	13.1 ± 1.0	222.1 ± 1.9	111.5 ± 2.0

Species in bold were analyzed in this study. Means of three replicates ± SD.

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
