# Peer review of "Edible Leafy Vegetables from West Africa (Guinea-Bissau): Consumption, Trade and Food Potential"

_foods, 2019, doi:10.3390/foods8100493_

Round 1

Reviewer 1 Report

The manuscript describes availability and some nutritional properties of several species of edible leafy vegetables in Guinea-Bissau. While it is certainly of interest, it must be noted that it gives hardly any new data that would be important for the wide audience. Also, it would be much more comprehensive if data from the wet season would be included. It is, however, relatively well written, quite easy to follow and presents the results that could interest some readers. Therefore, I suggest its acceptance after some (not very extensive) revision, including mostly more precise
description of the methods used. See below for specific comments.

1. Line 164: It is unclear how the extraction procedures were performed for whole leaves (djambos). Please specify, if they were used fresh, dried or lyophilized?
2. I am not fully convinced by the method applied for determination of the total phenolic content. It is not very popular and it might be a trouble to compare the results obtained in the present study with already published results, which were usually obtained using F-C method.
3. You report the results for the lipid content in Table 3, but there is no sign of description of the analytical method used in "Materials and methods" section. Please provide some details about it.
4. While majority of the results is in agreement with similar reports, the mineral content seems unusually high at least for some elements (K, Ca, Mg, P, Fe). This should be discussed a little bit more than it is in the current version of the manuscript.
5. Line 183: "as described in section 1.1" - there is no such section in the manuscript.
6. Table 1: "Ipomoea aquatica": please use italics.
7. Line 352: "Sesbania grandiflora": please use italics.
8. Lines 379-380: Please change: "it seems that there are room to" -> "it seems that there is some space" or similar.

Author Response

Reviewer 1 Comments

General comments:

The manuscript describes availability and some nutritional properties of several species of edible leafy vegetables in Guinea-Bissau. While it is certainly of interest, it must be noted that it gives hardly any new data that would be important for the wide audience. Also, it would be much more comprehensive if data from the wet season would be included. It is, however, relatively well written, quite easy to follow and presents the results that could interest some readers. Therefore, I suggest its acceptance after some (not very extensive) revision, including mostly more precise description of the methods used.

The authors greatly appreciate all the comments made by the reviewer. We believe that the data provided in this manuscript highlights the need to disclose nutritional quality of wild edible plants, particularly in developing countries as Guinea-Bissau, where food access is scarce as well as food diversification. To overcome food scarcity and meet dietary intakes, local populations have learnt to process plants to cope nutritional requirements, with djambos and lalos hereby represented in this paper as good mineral and protein sources for reducing food insecurity. Also, it emphasizes the need for more studies on Wild Edible Plants, particularly in developing countries where local populations have strong ethno-heritage knowledge of plants uses and benefits.

See below for specific comments.

Line 164: It is unclear how the extraction procedures were performed for whole leaves (djambos). Please specify, if they were used fresh, dried or lyophilized?

We acknowledge the reviewer comment and new text have been added to accommodate this information, and be read as follows at L153-154: “Fresh leaves (djambos) were lyophilized prior to chemical analysis.” and at L170:” Extracts from lyophilized fresh (djambos) and dried (lalos) leaves were (…)”.

I am not fully convinced by the method applied for determination of the total phenolic content. It is not very popular and it might be a trouble to compare the results obtained in the present study with already published results, which were usually obtained using F-C method.

Although we understand the Reviewer’s concern, the total phenolic content method used in our study is part of the determination of the antioxidant activity. We have used this method in previous works (e.g. Coelho et al., 2012, “Natural extracts from Pterospartum tridentatum at different vegetative stages: extraction yield, phenolic content and antioxidant activity”, https://core.ac.uk/download/pdf/62720214.pdf), with results acquired being similar to those obtained by C-F, and as such our data can be compared with already published results from C-F method.

You report the results for the lipid content in Table 3, but there is no sign of description of the analytical method used in "Materials and methods" section. Please provide some details about it.        

We appreciate the reviewer comment and in the revised version we have added the following sentence to include the analytical method used for lipid content analysis: L162- “Lipid content - The total lipid content was determined by the Soxhlet extraction with hexane as solvent.

While majority of the results is in agreement with similar reports, the mineral content seems unusually high at least for some elements (K, Ca, Mg, P, Fe). This should be discussed a little bit more than it is in the current version of the manuscript.

The authors thank the reviewer comment and have added new text in Discussion section including the high level some mineral elements obtained in extracts analyzed, and may read as follows at

L269-270: “Such high levels of minerals in non-fertilized crops could be related to soil and climate issues that should be further studied.”

L368-378: “High levels of several mineral elements, namely K, Ca, Mg, P, Fe, were observed in both djambos and lalos screened. Elevated levels of calcium in Adansonia digitata leaves (307 to 2640 mg/100 g dw) have also been reported [46], which is in accordance to our data. In a previous study, Leterme et al. [47] analysed the mineral content of 68 species from the tropical rainforest of Colombia, with several species displaying high contents in potassium (1.782mg/100g dw), calcium (280–1242 mg Ca/100 g edible portion) and iron (0.7–8.4 mg Fe/100 g edible portion). In fact, this study has reported for the first time the highest content of Ca on the leaves of an African species (Trichanthera sp., Acanthaceae), ca. 62 g Ca/kg DM, used in human nutrition. Considering the high mineral contents on lalos and djambos analyzed, one can recognized the great potential of these WEP to play a major role in a more sustainable and diversified diet, considering they can be regarded as good vegetable sources of many mineral elements.

 Line 183: "as described in section 1.1" - there is no such section in the manuscript.

We have considered this comment by the Reviewer and in therevised version of the manuscript and reads as follows, L189: “Lyophilized extracts of lalos and djambos (100 μL) (…)”.

Table 1: "Ipomoea aquatica": please use italics.

Line 352: "Sesbania grandiflora": please use italics.

All italics are now used as per reviewer indications.

Lines 379-380: Please change: "it seems that there are room to" -> "it seems that there is some space" or similar. The change has been done.

Reviewer 2 Report

The manuscript entitled Edible leafy vegetables from West Africa (Guinea Bissau): consumption, trade and food potential  presents the results of a analysis of some wild and semi-cultivated plants known as important food source in West Africa. The integration of the food plants mentioned in this study into the diet is beneficial due to their increased levels of minerals and biologically active compounds (e.g. phenolic derivatives) and also due to their remarkable antioxidant capacity.

The research design is appropriate, the conclusions are supported by the experimental results, and the paper is written according to the requirements imposed by the journal.

The manuscript is generally well written and it is easy to catch the essence. I am pleased to recommend the publication of this manuscript, owing to its interest and the careful way in which the data are presented.

Questions to be addressed:

Do you have any information about how they process the food starting materials, how they use the different plants as part of their diet? What do we know about the frequency of usage of each plant?

The mineral content especially the calcium and iron content seems to be surprisingly high. Do you have any information about the bioavailability of these minerals after consumption of the above mentioned edible plants? It is known that the processing method used can greatly influence the release of certain minerals from the plant matrix.

Author Response

Reviewer 2 comments

The manuscript entitled Edible leafy vegetables from West Africa (Guinea Bissau): consumption, trade and food potential presents the results of a analysis of some wild and semi-cultivated plants known as important food source in West Africa. The integration of the food plants mentioned in this study into the diet is beneficial due to their increased levels of minerals and biologically active compounds (e.g. phenolic derivatives) and also due to their remarkable antioxidant capacity.

The research design is appropriate, the conclusions are supported by the experimental results, and the paper is written according to the requirements imposed by the journal.

The manuscript is generally well written and it is easy to catch the essence. I am pleased to recommend the publication of this manuscript, owing to its interest and the careful way in which the data are presented.

The authors would like to thank the reviewer positive comments on the manuscript.

Questions to be addressed:

Do you have any information about how they process the food starting materials, how they use the different plants as part of their diet? What do we know about the frequency of usage of each plant?

We would like to thank the reviewer for the important question. Data remains scarce regarding the frequency of consumption of these food products, but as far as we know they are consumed often but in small quantities, in particular the lalos. Considering that lalos, and to less extent, djambos are used in daily food preparations namely in rice and stews, the frequency of consumption can be assumed as daily basis. To accommodate the reviewer comment, we have included information regarding the frequency of usage of the plants, and may read as follows in the revised manuscript:

L344-347- “The availability of these products throughout the year is relatively constant: lalos are dried and ground during the species' leaf production season and preserved throughout the year, while djambos’ species can be produced during the year as they are cultivated, thus watered, and consumed as fresh leaves.”

The mineral content especially the calcium and iron content seems to be surprisingly high. Do you have any information about the bioavailability of these minerals after consumption of the above mentioned edible plants? It is known that the processing method used can greatly influence the release of certain minerals from the plant matrix.

Regarding the bioavailabilty of these minerals after consumption, we do not have much information, and further studies should now be focused on the analysis of the processing method of both djambos and lalos and its effect on the minerals availability/release.

Also, we have included new text in the Discussion section to accommodate comments raised by the reviewer on the high mineral contents of calcium and iron, as can be read as follows in the revised manuscript:

L368-378: “High levels of several mineral elements, namely K, Ca, Mg, P, Fe, were observed in both djambos and lalos screened. Elevated levels of calcium in Adansonia digitata leaves (307 to 2640 mg/100 g dw) have also been reported [46], which is in accordance to our data. In a previous study, Leterme et al. [47] analysed the mineral content of 68 species from the tropical rainforest of Colombia, with several species displaying high contents in potassium (1.782mg/100g dw), calcium (280–1242 mg Ca/100 g edible portion) and iron (0.7–8.4 mg Fe/100 g edible portion). In fact, this study has reported for the first time the highest content of Ca on an African tree leaves (Trichanthera sp., Acanthaceae), ca. 62 g Ca/kg DM, used in human nutrition. Considering the high mineral contents on lalos and djambos analyzed, one can recognized the great potential of these WEP to play a major role in a more sustainable and diversified diet, considering they can be regarded as good vegetable sources of many mineral elements.